# Rapid Identification of Wild *Gentiana Genus* in Different Geographical Locations Based on FT-IR and an Improved Neural Network Structure Double-Net

**DOI:** 10.3390/molecules27185979

**Published:** 2022-09-14

**Authors:** Pan Zeng, Xiaokun Li, Xunxun Wu, Yong Diao, Yao Liu, Peizhong Liu

**Affiliations:** 1School of Medicine, Huaqiao University, Quanzhou 362021, China; 2School of Biomedical Science, Huaqiao University, Quanzhou 362021, China; 3College of Science and Engineering, National Quemoy University, Kinmen 89250, Taiwan; 4College of Engineering, Huaqiao University, Quanzhou 362021, China

**Keywords:** wild *Gentiana Genus*, FT-IR spectroscopy, deep learning, Double-Net, geographical location identification

## Abstract

*Gentiana Genus*, a herb mainly distributed in Asia and Europe, has been used to treat the damp heat disease of the liver for over 2000 years in China. Previous studies have shown significant differences in the compositional contents of wild *Gentiana Genus* samples from different geographical origins. Therefore, the traceable geographic locations of the wild *Gentiana Genus* samples are essential to ensure practical medicinal value. Over the last few years, the developments in chemometrics have facilitated the analysis of the composition of medicinal herbs via spectroscopy. Notably, FT-IR spectroscopy is widely used because of its benefit of allowing rapid, nondestructive measurements. In this paper, we collected wild *Gentiana Genus* samples from seven different provinces (222 samples in total). Twenty-one different FT-IR spectral pre-processing methods that were used in our experiments. Meanwhile, we also designed a neural network, Double-Net, to predict the geographical locations of wild *Gentiana Genus* plants via FT-IR spectroscopy. The experiments showed that the accuracy of the neural network structure Double-Net we designed can reach 100%, and the F1_score can reach 1.0.

## 1. Introduction

*Gentiana Genus* is an herb found mainly in Asia and Europe that has been used for treating liver damp heat disease for more than 2000 years in China [1]. Recent studies have demonstrated the efficacy of Gentiana in treating diabetes, with liver protection and anti-inflammatory properties [2,3,4,5]. The TCM classic herbal formula with Gentiana as the ‘Jun herb’ (critical herb), Longdanxiegan, has been widely prescribed in treating hypertension by TCM physicians in China [6]. The literature data on the chemical composition suggests that the medicinal substances of Gentiana are iridoids (gentiopicroside, swertidine), flavonoids (isoorientin), xanthones, and polysaccharides [7,8,9]. Differing from chemical drugs, herbal products exhibit their curative efficacy on the basis of multi-components and multi-targets, and the medicinal substances are closely related to the soil, climate, harvest season, growth age, and other factors [10,11]. As an important medicinal plant, Gentiana is widely distributed in the temperate mountainous regions of China [12]. Previous studies have shown significant differences in the contents of wild Gentiana samples from different geographical sources [13,14]. Hence, tracing the geographical origin of wild Gentiana samples is crucial to ensure their valid medicinal value, which will help to ensure the potency of the herb.

Several advanced spectral and chromatographic techniques have been successfully utilized to identify the authenticity and quality of various herbal medicines, including Fourier transform infrared (FT-IR), high-performance liquid chromatography (HPLC), ultra-performance liquid chromatography (UPLC), nuclear magnetic resonance (NMR), and Raman spectroscopy techniques [15,16,17]. However, some of the above methods require complicated sample pre-processing procedures and generate a considerable amount of organic solvent waste solution during the experiment. In contrast, FT-IR spectroscopy is more widely used because of its advantage of allowing rapid and nondestructive measurements [18,19]. Furthermore, the sample volume required for FT-IR testing is very small (down to a few milligrams). As a result, it is more sensitive and possibly more appropriate to obtain valuable information from the sample profile. FT-IR spectroscopy focuses on the MIR region of the electromagnetic spectrum, which could provide information about the foundational vibration (from the stable vibrational state to the first excited vibrational state) of the chemical functional group [20].

With the rise of machine learning and deep learning, more and more people are using machine learning and deep learning algorithms to accomplish scientific research tasks related to spectra. Zareef et al. used an improved machine science collaborative interval partial least squares algorithm combined with competitive adaptive reweighted sampling (Si-CARS-PLS) to predict the antioxidant activity of walnuts with good results [21]. The prediction of gentian from various geographical sources using high-performance liquid chromatography (HPLC) and Fourier transform infrared spectroscopy (FT-IR) was also achieved by Zhao et al. [22]. In the work by Wu et al. [13], support vector machines (SVM) and the related PLS algorithm were used in combination with FT-IR spectroscopy and HPLC to evaluate the quality of wild Gentiana rigescens from different regions, and it was shown experimentally that the improved PLS and SVM algorithms can be used as an alternative method for the qualitative identification and quantitative evaluation of the quality of Gentiana rigescens. In the work by Pei et al. [23], they fused mid-infrared (MIR) and near-infrared (NIR) data and used random forest (RF) and partial least squares discriminant analysis (PLS-DA) machine learning models to predict wild *Paris polyphylla* populations in Yunnan. They also used a principal component analysis (PCA) and certain algorithms for the essential feature selection to extract important features, and finally also achieved a good experimental result with 100% accuracy. There has also been plenty of related work using machine learning algorithms [24,25].

In addition to machine learning algorithms [26], deep learning algorithms [27,28], such as artificial neural networks, also show better performance. There have also been many better research studies using deep learning models to process data related to spectra. In the work by Mutlu et al. [29], FT-IR spectra were used to predict the quality parameters of wheat, and they performed a chemical analysis of flour samples from 79 different wheat varieties grown in different regions of Turkey and used NIR spectra to train the artificial neural network (ANN), and finally achieved a better result. In the work by Gonzalez-Viejo et al. [30], they used PLS and ANN correlation algorithms to predict four chemometrics of beer, such as the pH, alcohol, brix, and maximum volume of foam. After the experiments, they showed that the artificial neural networks could predict these four chemometrics well and that the R^2^ of their model built with neural networks reached 0.95. Neural network models have been used in many scientific tasks with good results, and in addition to the application of neural networks in FT-IR spectroscopy, there are many applications of neural networks in other fields. For example, Qie et al. used a BP neural network to study the trajectory planning of redundant robotic arms during upper limb rehabilitation [31], and they demonstrated the feasibility of their method through relevant experiments. Fatigue driving has been a hot topic for a long time, and Chen et al. combined a BP neural network with a time-cumulative effect to detect drowsy driving [32]. They used three features related to fatigue (the longest time a driver closes their eyes continuously, the number of yawns in a period, and the time their eyes are closed) as inputs to the neural network, and finally built an effective model for driver fatigue detection. 

The aim of this study was to build a fast, nondestructive, and efficient method for the identification of the geographical origin of wild *Gentiana Genus* using FT-IR combined with chemometrics. Meanwhile, for the identification of the geographic location of the wild *Gentiana Genus*, we designed a well-performing neural network structure, Double-Net. Due to its good performance, we believe that Double-Net can be applied for the geographical origin identification of other similar herbs.

## 2. Results

### 2.1. Results of Data Pre-Processing

#### 2.1.1. Pre-Processing Results for FT-IR Spectroscopy 

The results of the different pre-processing methods for the FT-IR spectra are shown in Figure 1. For the pre-processing of wild *Gentiana Genus* FT-IR spectral data, in addition to using a single pre-processing method in Figure 1, we also combined different pre-processing methods in our experiments to observe the experimental results of the different models.

#### 2.1.2. Results of PCA Processing 

The percentages of each of the 20 principal components in the FT-IR spectra processed in six different ways are shown in Figure 2. It can be seen from Figure 2 that when we used 20 principal components, the first 15 principal components had the largest contribution and the last few principal components contributed almost 0 (variance ratio < 0.1%). In order to make the machine learning model, we used more features for learning and we input all of the 20 obtained principal components into the model.

### 2.2. Macroscopic Chemistry Components in IR Spectra 

The roots of gentian are rich in iridoids (gentiopicroside, swertiamarin, loganin), flavonoids (luteolin, isoorientin, apigenin, and chrysoeriol), xanthones (mangiferin, gentisin, and its glycosides), and polysaccharides [15,16,17]. The raw FT-IR spectrum of one of the wild *Gentiana Genus* samples is shown in Figure 3. The 2500–3700 cm^−1^ region is called the hydrogen stretching zone, as the vibration frequencies of C-H, N-H, and O-H appear in this area. The figure shows that the first C-H stretching vibration peaks appear at 2923 and 2850 cm^−1^. The region of 2000–2300 cm^−1^ is referred to as the triple bond stretching region (C≡C and C≡N), with almost no peaks in the IR spectrum of the Gentiana sample. The 1600–2000 cm^−1^ region is known as the double bond stretching region (C=C, C=N, and C=O). The peaks around 1736 cm^−1^ represent the CO stretching vibration of the ester, and the peak at 1615 cm^−1^ suggests the presence of free carboxyl groups or carbohydrates. The peaks at 1421 and 1375 cm^−1^ represent the asymmetric bending vibration of the methyl group, which is the result of the esters. The presence of intense bands at 1025 cm^−1^ is considered to be caused by the glucose skeleton. The last interval in the FT-IR spectra is the fingerprint region, from 1300 to 400 cm^−1^, which could exhibit more detailed functional group information for the sample [33]. The spectrograms of Gentiana samples from different regions were very similar, so we needed to rely on chemometrics for the analysis [34].

### 2.3. Dataset Description

In terms of deep learning, since the deep learning model can go through many features to select those that are important for the current task, there is no need for us to manually extract the sample features artificially, so for the BP neural network and our own designed neural network Double-Net, the data we input for each sample are all the features of the FT-IR spectrum.

In order to make the model show the best performance and also to better test the performance of the model, 20% of the wild *Gentiana Genus* data are divided into the test data set, and 80% are divided into the training data set. Meanwhile, to make the data for the wild *Gentiana Genus* more reasonable, we used stratified sampling for FT-IR spectral data of the wild *Gentiana Genus* according to the different locations. The first column shows the different pre-processing methods for FT-IR spectra, where NO_OP indicates the input for the raw FT-IR data (without any pre-processing of FT-IR spectra).

### 2.4. Models Verification

The specific accuracy and F1_score values for different pre-processing methods for FT-IR spectral data of wild *Gentiana Genus* genera are shown in Table 1. Acc in the table denotes accuracy, and F1_score in the table denotes the F1_score of the model. The first column of the table shows the different pre-processing methods for FT-IR spectra of the wild *Gentiana Genus*, and the NO_OP means that the FT-IR spectral data input to the model are raw (without any pre-processing of the FT-IR spectra).

#### 2.4.1. Machine Learning Models

As presented in Table 1, the highest accuracy of the decision tree model for predicting the geographic location of the wild *Gentiana Genus* was 84.44%, and the highest F1_score was 0.81; the highest accuracy of the naive Bayes model for predicting the geographic location of the wild *Gentiana Genus* was 91.10%, and the highest F1_score was 0.91; the highest accuracy of the SVM model for predicting the geographic location of the wild *Gentiana Genus* was 83.3%, and the highest F1_score was 0.91. The accuracy and F1_score values of these machine learning models with different pre-processing methods are shown in Figure 4. Since the FT-IR spectral data we obtained had some noise, after using different spectral pre-processing methods, we achieved different degrees of denoising. For decision tree models, Norm + WT is a better pre-processing method; for the naive Bayes model, WT + MSC and MC + WT are the best pre-processing methods to make the model perform better. For the SVM model, the best pre-processing method is WT + SNV.

#### 2.4.2. BP Neural Network

As shown in Figure 4d and Table 1, the BP neural network model predicted the geographic location of the wild *Gentiana Genus* with the highest accuracy value of 97.78% and the highest F1_score of 0.97. For the BP neural network, the effective pre-processing methods for FT-IR spectral data of the wild *Gentiana Genus* were SNV, Norm + WT, SG + MC, and MSC + SNV. Compared to the machine learning models, the BP neural networks improved in performance. However, there was a decrease in the BP neural network performance for some pre-processing data methods (e.g., MSC, Norm + MSC, and MC + MSC). 

#### 2.4.3. Double-Net

Based on the performance of the BP neural network in predicting the geographic location of the wild *Gentiana Genus*, we believe that there is much potential for improvement, and after many experiments, we designed a neural network structure called Double-Net that performs better. As shown in Figure 5 and Table 1, we designed the neural network Double-Net for the geographic location prediction of the wild *Gentiana Genus* with the highest accuracy of 100% and the highest F1_score of 1.0. Based on our experiments, for different methods of data pre-processing, the average accuracy of Double-Net can reach 94.48% and the average F1_score can reach 0.97. Compared with the models used in this experiment, the neural network structure Double-Net that we designed performs better than other models for various data processing methods. This proves the effectiveness of our designed neural network structure Double-Net. 

For the neural network Double-Net that we designed, there are 8 FT-IR spectral pre-processing methods that make the model perform the best. These eight methods are SG, SNV, Norm + WT, Norm + SNV, SG + WT, MC + SNV, WT + MSC, and WT + SNV. This means that the pre-processing methods are essential for the FT-IR spectral data. It was also found that our neural network Double-Net performs better in predicting the geographic location of the wild *Gentiana Genus* by pre-processing the FT-IR spectral data. In addition, we also found from Table 1 that the accuracy of both the BP neural network and Double-Net can reach more than 95% without FT-IR spectral pre-processing. There are 7468 features being fed into the deep learning model (far more than the number of features fed into the machine learning models). There is no doubt that there are many unimportant features among the 7468 features, which means that the deep learning model needs to select the major features among the 7468 features to predict the geographic location of the wild *Gentiana Genus*. This is a powerful reflection of the advantage of deep learning models, which can select the significant features among many features to achieve better performance.

## 3. Discussion

In previous studies [13,18,19,20,21,22], most of the methods used to accomplish herb location prediction were machine learning algorithms. Machine learning methods (e.g., PLS, SVM) work well for data with few sample features. However, when there are more features in the sample to be processed, the following solutions are commonly used. First, based on human a priori knowledge, the important features are selected among many features and then processed via machine learning algorithms. Second, high-dimensional data are first changed into easy-to-process low-dimensional data by using a dimensionality reduction algorithm, and then processed using certain machine learning methods. Third, an important feature selection algorithm is used to select certain critical features, which to a certain extent achieves the dimensionality reduction, and then certain machine learning algorithms are used for processing. All of these algorithms contain a tedious step, meaning they have to find certain crucial features among the many features to perform the following steps. 

However, with the improvement of scientific research, the algorithms related to deep learning have performed well in many fields and solved many challenges. Among the many deep learning algorithms, combined with the relevant data for this experiment, the neural network is a very applicable network model. A neural network is an algorithm that simulates the human brain, and the use of activation functions such as ReLU, Tanh, and Sigmoid in the neural network can mean the relevant model of the neural network has stronger nonlinear characterization ability [35]. When using an artificial neural network, we can hand over all of the features of the sample to the artificial neural network, which removes the need to manually extract the relevant variable features of the sample. All features of the sample are input into the model of the artificial neural network, which also enables the artificial neural network to have more features to learn, so that the model has a better basis for making decisions. At the same time, this can also avoid the occurrence of poor model results due to feature extraction errors, so that the model can achieve better results. For the task of predicting the geographic location of the wild *Gentiana Genus* using FT-IR spectra, because all data from the FT-IR spectra of the wild *Gentiana Genus* were input into the deep learning model, such that the deep learning model was able to make full use of the relevant information in the TF-IR spectral data of the wild *Gentiana Genus*, which is perhaps one of the reasons for the improved performance of the deep learning model.

Differing from chemical drugs, herbal products have multi-component and multi-target efficacy, and the resulting drugs are closely related to the soil, climate, harvesting season, and growing age, among other factors. Meanwhile, these factors also affect the contents of herbal ingredients. The information related to specific chemical functional groups (from the stable vibrational state to the first excited state) can be found in FT-IR spectra. For instance, the information related to groups such as C-H, O-H, N-H, and C=O is able to be reflected in the FT-IR spectra. In order to establish a rapid, harmless, efficient, and combined chemometric method to identify the geographic origin of the wild *Gentiana Genus*, we collected wild *Gentiana Genus* herbs from seven provinces, and we obtained the FT-IR spectral data of wild Gentiana using FT-IR spectroscopy instruments. Finally, we designed a neural network called Double-Net that performed well on the wild *Gentiana Genus* dataset, and the network performed well in the geographic location prediction of the wild *Gentiana Genus*. 

After relevant experiments, we validated the use of FT-IR spectral information to efficiently, rapidly, and nondestructively predict the geographic location of the wild *Gentiana Genus* as a workable solution. Given the excellent performance of Double-Net, we believe that this neural network model we designed can be used to roughly identify the geographical location of the wild *Gentiana Genus*. In addition to the wild *Gentiana Genus*, we believe that Double-Net can also be used for the geographical location identification of other medicinal herbs. The use of FT-IR spectroscopy data enables the nondestructive and rapid identification of the geographical locations of herbs, which is a good way to avoid time-consuming, laborious, and costly losses of herbs. This is good news for the geographical identification of valuable herbs. Moreover, we believe that more work like this can be carried out in the future based on our experiments.

## 4. Materials and Methods

### 4.1. Samples Preparation

The 222 roots of wild *Gentiana Genus* samples were collected from seven geographical locations in P.R China (Jiangxi, Sichuan, Yunnan, Guizhou, Hubei, Hunan, and Shanxi), as shown in Figure 6a, and the distribution of the number of wild *Gentiana Genus* samples at each location is shown in Figure 6b. All wild samples were identified as *Gentiana Genus* by Professor Xianxiang Xu (School of medicine, Huaqiao University). All root samples were washed with tap water and were dried in a drying oven at 50 °C, then milled and sifted through 80 mesh sieves. All samples were packed in polyethylene zip-lock bags and stored in a dry environment for a further analysis. 

### 4.2. Fourier Transform Infrared (FT-IR) Spectroscopy Analysis

The infrared absorption spectra of samples were recorded using an FT-IR spectrometer equipped with a deuterated triglycine sulfate (DTGS) detector and an ATR (attenuated total reflection) accessory (NICOLET iS 50, Thermo Fisher, Waltham, MA, USA). Typically, the accumulation spectra of 16 scans per sample were collected and averaged. The absorption spectra in the area between 4000 and 400 cm^−1^ with a 4 cm^−1^ resolution were obtained. Three analytical replicates of FT-MIR spectral data of all wild *Gentiana Genus* samples were obtained.

### 4.3. Data Pre-Processing 

#### 4.3.1. Raw Spectrum and Its Processing

The raw FT-IR spectra of the wild *Gentiana Genus* samples are presented in Figure 7, and the FT-IR spectra of the wild *Gentiana Genus* from different regions do not vary much. For some classification models, particularly for machine learning models, it is difficult to find subtle differences to distinguish the wild *Gentiana Genus* samples from different areas. In order to improve the robustness and performance of the model we built, we needed to use some data enhancement algorithms to process the raw data before we built the model, such as removing noise from the data, reducing random errors in the data, and eliminating baseline drift interference. For data processing, we used the data processing methods used by Rinnan et al. and Shao et al. in the processing of the FT-IR spectra, such as normalization (Norm), Savitzky–Golay (SG), and wavelet transform (WT) methods, to pre-process our wild *Gentiana Genus* spectral data [36,37].

#### 4.3.2. Exploratory Analysis of PCA

Since the machine learning models used in our experiments require the manual extraction of some features to be input to the models, in order to make our input to the machine learning models more objective and make our machine learning models perform better, we used the principal component analysis (PCA) algorithm to reduce the dimensionality of the 7468-dimensional data. 

### 4.4. Models

#### 4.4.1. Machine Learning Models

In the experiments, to evaluate the models more objectively, we used models with good results in performance sharing among machine learning methods [38], such as decision trees, plain Bayesian, and support vector machine (SVM) classification models, and compared them with deep learning algorithms and BP neural networks that have good performance in classification. The decision tree is a tree structure built using the entropy of information or the Gini index. For the input features of the decision tree, the more important features will be distributed at the top of the tree structure, and the decision tree model will judge these important input features first and then the other less important input features. After the decision tree model, each sample can be assigned to a specific category. The naive Bayes classification model has a long history and is widely used for spam filtering and news classification. Of course, the naive Bayes classification algorithm can be used not only for the classification of textual data, but also for different classification tasks depending on the target task. In the case of an input sample, the naive Bayes model will use all input features. In prediction, it will predict the probability for each class in the current classification task, and in the prediction result the current sample is predicted for the class with the highest classification probability. The support vector machine algorithm is a supervised learning algorithm. For simple binary classification data, the SVM can learn a maximum-margin hyperplane to achieve the classification of binary data. Meanwhile, the SVM can also classify the data nonlinearly using the kernel method. Based on the principle of the binary classification of the data via SVM, the SVM can also perform certain multiclassification tasks using pairwise classification methods. 

#### 4.4.2. BP Neural Networks

Artificial neural networks (ANN), also known as neural networks (NN), are types of algorithms that mimic the structure and function of biological neural networks. There are three neural network layers (input, output, and hidden layers) in a classical neural network. The structure of a classical neural network is shown in Figure 8a, where each blue circle represents a neuron, which is also called a perceptron, and the neurons in a neural network are an imitation of the neurons in the human brain. The structure of a neuron in a neural network is shown in Figure 8b, where $X_{1},\;X_{2},\ldots ,\;X_{n}$ represents the input to the neuron and $W_{1},\;W_{2},\;\ldots ,W_{n}$ represents the weight of the current neuron on these inputs to $X_{1},\;X_{2},\ldots ,\;X_{n}$; b represents the bias; $F\left(u\right)$ represents the activation function (used to complete the nonlinear transformation of the data); *u* represents the output of the neuron when it is not activated by the activation function; and Y represents the final output of the neuron. The calculation of these variables is shown in Equations (1) and (2). The data from the neurons in the hidden layers and the output layer of a neural network are calculated from the input data and then they are output. The neurons in a neural network are connected between layers but not connected between neurons in the same layer of the neural network. The final layer, the output layer, is also usually called the fully connected layer.
(1)$$u=\overset{n}{\underset{i=1}{\sum }}\left(W_{i}\;*\;X_{i}+b\right)$$
(2)$$Y=F\left(u\right)$$

BP neural networks usually have multiple layers of perceptrons between the input and output layers [39], which can fit any linear and nonlinear function. The concept of a neural network was proposed by scientists led by Rumelhart and McClelland in 1986, which was back-propagated according to the error (usually using a gradient descent algorithm to optimize the network). Together with training on a given dataset, it is possible to optimize a BP neural network that can be used for different tasks. 

#### 4.4.3. Improved Neural Network Structure (Double-Net)

In order to build a good model for predicting the geographic location of the wild *Gentiana Genus*, we used algorithms with good performance in machine learning, such as decision tree, naive Bayes, and support vector machine methods. After experimentation, these models use the FT-IR spectral data of the wild *Gentiana Genus* to predict the geographic location of the wild *Gentiana Genus*. When the sample data contain a large number of features, the processing performance during deep learning will be better. BP neural networks have achieved good results in many fields through their excellent performance, especially in classification tasks. Considering that every wild *Gentiana Genus* sample has 7468 spectral datapoints (the step size is 0.5), we used the BP neural network to complete the prediction of the geographic location of the wild *Gentiana Genus*. The BP neural network greatly enhanced the performance of predicting the geographic location of the wild *Gentiana Genus*, but the BP neural network did not achieve the desired effect. Consequently, we tried to design a neural network structure to predict the geographical location of the wild *Gentiana Genus*. 

Inspired by the network structure of the Siamese neural network [40], we designed a neural network structure (Double-Net) with exceptional performance in the task of predicting the geographic location of the wild *Gentiana Genus*. Using our designed neural network structure Double-Net, the accuracy of predicting the geographic locations of wild *Gentiana Genus* samples using FT-IR spectroscopy can reach 100%, and our neural network Double-Net structure is shown in Figure 9, where the blue and yellow circles represent the neurons of the neural network, the red circle indicates the summation operation of the parameters of the two neural networks, $X_{1},X_{2},\;\ldots ,\;X_{n}$ indicates the model of the input (FT-IR spectral data of wild *Gentiana Genus* samples), and Y indicates the output of the model (geographic location of the wild *Gentiana Genus*).

### 4.5. Evaluation of the Model Performance

We applied multiple model evaluation metrics to evaluate the performance of the models. For the evaluation of the classification task models, the metrics used are usually the precision, recall, F1_score, and accuracy of the model prediction. Through the precision results, we can judge the performance of the model classification process. The closer the accuracy is to 100%, the more efficient the model is. The recall rate indicates whether the model can classify positive samples as positive samples, reflecting the capacity of the model to distinguish between each type of sample. The calculation formulas for precision and recall are shown in Equations (3) and (4), where TP (true positive) indicates the number of correct predictions by the model for the true location of the sample. FP (false positive) represents the number of incorrect predictions by the model for the location of the negative sample, whereby the sample is incorrectly predicted as a positive sample from a certain location, but the sample is actually a sample from another location. TN (true negative) is the number of negative samples that the model predicts properly. FN (false negative) represents the number of errors in the model’s prediction of positive samples, whereby the sample is predicted to be a positive sample from other locations, but the sample is actually a positive sample from the current location:(3)$$\mathrm{Precision}=\frac{\mathrm{TP}}{\mathrm{TP}+\mathrm{FP}}$$
(4)$$\mathrm{Recall}=\frac{\mathrm{TP}}{\mathrm{TP}+\mathrm{FN}}$$

The F1_score is a comprehensive metric of the precision and recall, which is a better metric of the model performance than the precision and recall, so we adopted the F1_score as one of the metrics of our model (the formula of F1_score is Equation (5). Furthermore, the accuracy is also a measure of the model performance for classification models, and the accuracy indicates the number of correct predictions among all samples (including positive and negative samples). The accuracy is used in our model evaluation metrics, and the calculation of the accuracy is shown in Equation (6):(5)$$F1\_\mathrm{Score}=\frac{\mathrm{Precison}\;*\;\mathrm{Recall}}{2*\left(\mathrm{Precision}\;+\;\mathrm{Recall}\right)}$$
(6)$$\mathrm{Accuracy}=\frac{\mathrm{TP}+\mathrm{TN}}{\mathrm{TP}+\mathrm{FP}+\mathrm{TN}+\mathrm{FN}\;}$$

### 4.6. Software

This experiment was based on the Windows 10 operating system. The FT-IR spectra were processed using Omnic (Version 8.2, Thermo Fisher Scientific, Madison, WI, USA). All models were created using PyCharm (version 2021 professional), and the pre-processing of the FT-IR data was also done using PyCharm (version 2021 professional). The programming environment was Python 3.7, and the deep learning framework used in our study was PyTorch 1.7.

## 5. Conclusions

In this study, the geographical location of the wild *Gentiana Genus* was predicted using benchtop FT-IR spectroscopy coupled with an improved neural network structure Double-Net. Here, 21 FT-IR spectral data pre-processing methods and 5 efficient algorithms were used for comparison and evaluation. The experiments showed that our improved neural network structure, Double-Net, is the optimal model for predicting the geographic locations of wild *Gentiana Genus* samples. Our improved neural network structure, Double-Net, achieved 100% accuracy and an F1_score of 1.0 on the test dataset of the wild *Gentiana Genus*. This means that it can be used to establish a rapid, nondestructive, and efficient method for the identification of the geographical locations of wild *Gentiana Genus* plants combined with chemometrics. Given that this experiment is a preliminary study, we believe that FT-IR spectroscopy can be used to explore the geographical locations of more traditional Chinese medicines in the future.

## Figures and Tables

**Figure 1 molecules-27-05979-f001:**
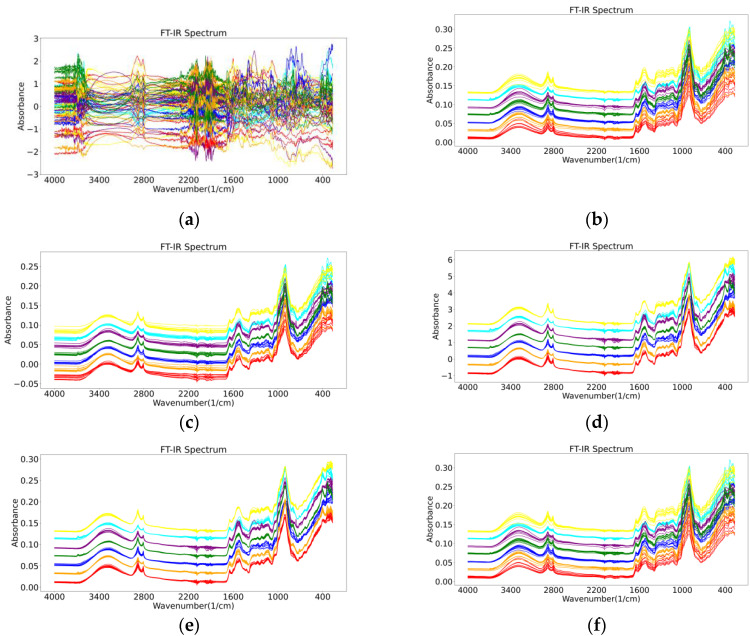
The results of six different pre-processing methods for FT-IR spectra of wild *Gentiana Genus*: (**a**) normalization (Norm); (**b**) Savitzky–Golay (SG); (**c**) mean centralized (MC); (**d**) standard normalized variate (SNV); (**e**) multivariate scatter correction (MSC); (**f**) wavelet transform (WT).

**Figure 2 molecules-27-05979-f002:**
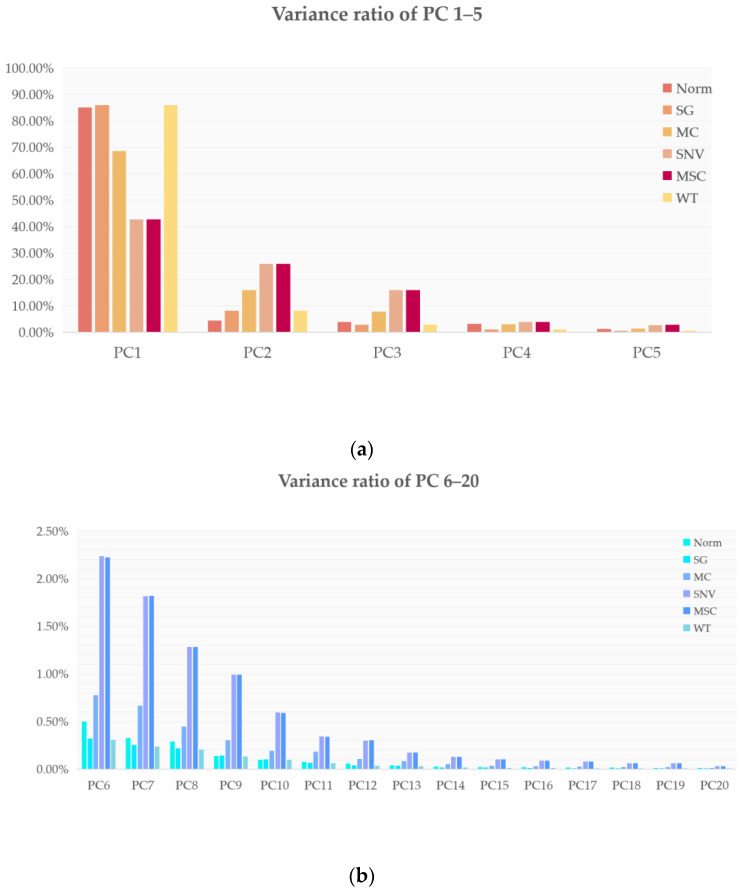
Twenty principal component contributions for six different data pre-processing methods: (**a**) percentage contributions of the 1st to 5th principal components for the six different data pre-processing methods; (**b**) percentage contributions of the 6th to 20th principal components for the 6 different data pre-processing methods.

**Figure 3 molecules-27-05979-f003:**
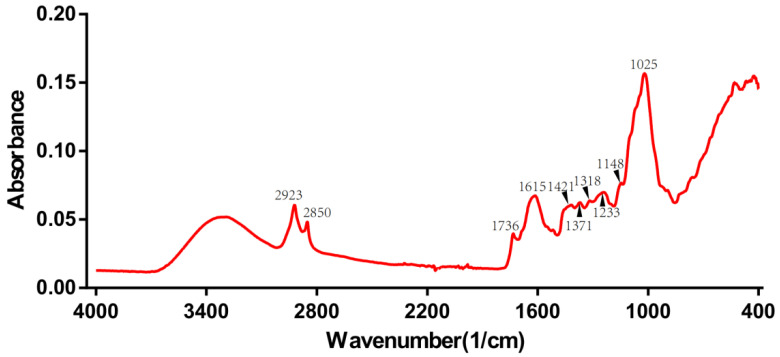
The raw FT-IR spectrum of one of the wild *Gentiana Genus* samples.

**Figure 4 molecules-27-05979-f004:**
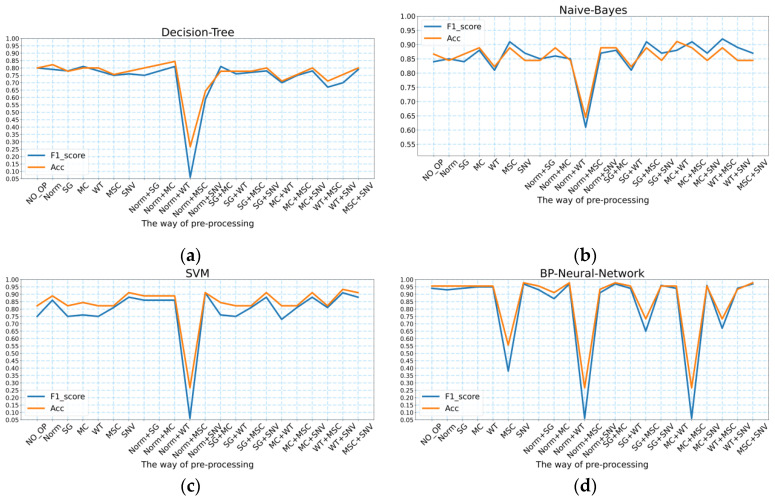
The accuracy and F1_score values of the four models under different pre-processing methods: (**a**) decision tree; (**b**) naive Bayes; (**c**) SVM; (**d**) BP neural network.

**Figure 5 molecules-27-05979-f005:**
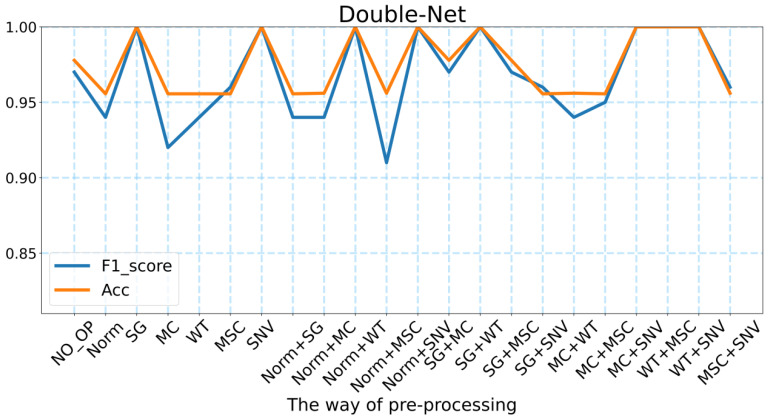
The accuracy and F1_score of Double-Net under different pre-processing methods.

**Figure 6 molecules-27-05979-f006:**
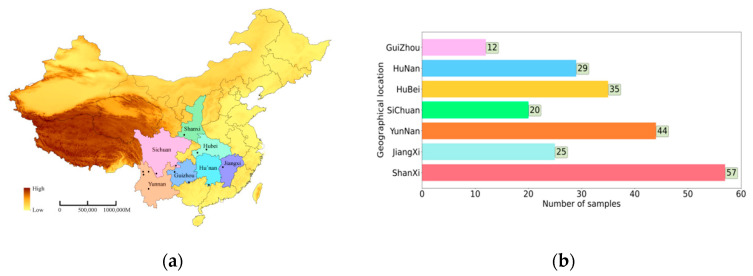
The data sources and data distribution: (**a**) seven geographic sampling locations of wild *Gentiana Genus* samples; (**b**) the distribution of wild *Gentiana Genus* samples in 7 provinces.

**Figure 7 molecules-27-05979-f007:**
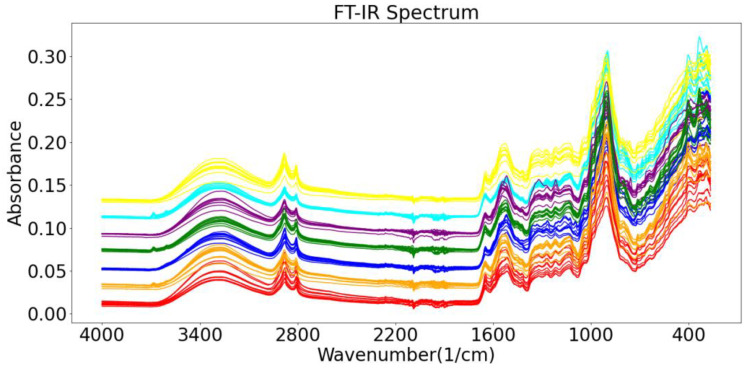
Raw FT-IR spectral data of wild *Gentiana Genus*.

**Figure 8 molecules-27-05979-f008:**
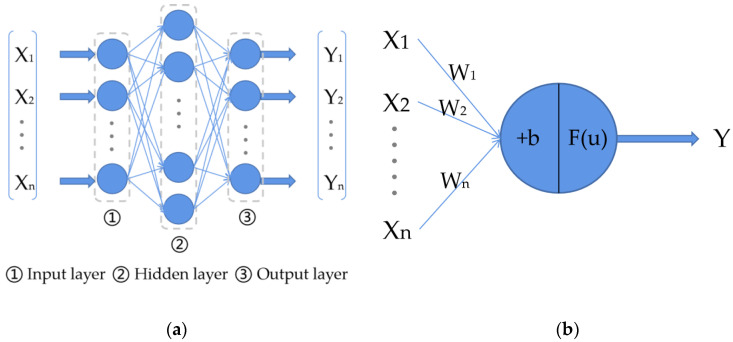
The structure of a classical neural network and a neuron: (**a**) classical neural network with three layers; (**b**) the structure of a neuron in a neural network.

**Figure 9 molecules-27-05979-f009:**
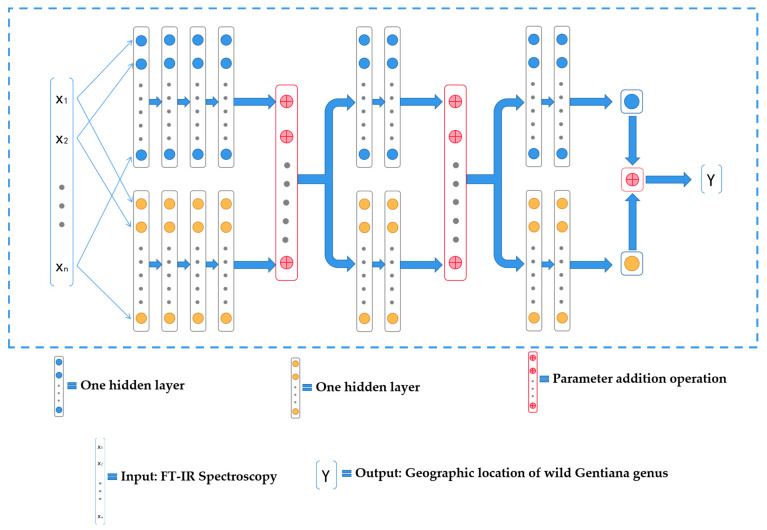
The structure of the neural network Double-Net.

**Table 1 molecules-27-05979-t001:** Performance of each model.

Model	Decision Tree		Naive Bayes		SVM		BP Neural Network		Double-Net (Ours)	
Evaluation Metrics	Acc	F1_Score	Acc	F1_Score	Acc	F1_Score	Acc	F1_Score	Acc	F1_Score
NO_OP	80.00%	0.80	86.67%	0.84	82.20%	0.75	95.56%	0.94	97.78%	0.97
Norm	82.22%	0.79	84.44%	0.85	88.90%	0.86	95.56%	0.93	95.56%	0.94
SG	77.78%	0.78	86.67%	0.84	82.20%	0.75	95.56%	0.94	**100.00%**	**1.00**
MC	80.00%	0.81	88.89%	0.88	84.40%	0.76	95.56%	0.95	95.56%	0.92
WT	80.00%	0.78	82.22%	0.81	82.20%	0.75	95.56%	0.95	95.56%	0.94
MSC	75.56%	0.75	88.89%	0.91	82.20%	0.81	55.56%	0.38	95.56%	0.96
SNV	77.78%	0.76	84.44%	0.87	91.10%	0.88	**97.78%**	**0.97**	**100.00%**	**1.00**
Norm + SG	80.00%	0.75	84.44%	0.85	88.90%	0.86	95.56%	0.93	95.56%	0.94
Norm + MC	82.22%	0.78	88.89%	0.86	88.90%	0.86	91.11%	0.87	95.60%	0.94
Norm + WT	**84.44%**	**0.81**	84.44%	0.85	88.90%	0.86	**97.78%**	**0.97**	**100.00%**	**1.00**
Norm + MSC	26.67%	0.06	64.44%	0.61	26.70%	0.06	26.67%	0.06	95.60%	0.91
Norm + SNV	64.44%	0.59	88.89%	0.87	91.10%	**0.91**	93.33%	0.91	**100.00%**	**1.00**
SG + MC	77.78%	0.81	88.89%	0.88	84.40%	0.76	**97.78%**	**0.97**	97.78%	0.97
SG + WT	77.78%	0.76	82.22%	0.81	82.20%	0.75	95.56%	0.94	**100.00%**	**1.00**
SG + MSC	77.78%	0.77	88.89%	0.91	82.20%	0.81	73.33%	0.65	97.78%	0.97
SG + SNV	80.00%	0.78	84.44%	0.87	91.10%	0.88	95.56%	0.96	95.56%	0.96
MC + WT	71.11%	0.70	**91.10%**	0.88	82.20%	0.73	95.56%	0.94	95.60%	0.94
MC + MSC	75.56%	0.75	88.89%	0.91	82.20%	0.81	26.67%	0.06	95.56%	0.95
MC + SNV	80.00%	0.78	84.44%	0.87	91.10%	0.88	95.56%	0.96	**100.00%**	**1.00**
WT + MSC	71.11%	0.67	88.89%	**0.92**	82.20%	0.81	73.33%	0.67	**100.00%**	**1.00**
WT + SNV	75.56%	0.70	84.44%	0.89	**93.30%**	**0.91**	93.33%	0.94	**100.00%**	**1.00**
MSC + SNV	80.00%	0.79	84.44%	0.87	91.10%	0.88	**97.78%**	**0.97**	95.60%	0.96
Max	84.44%	0.81	91.10%	0.92	93.30%	0.91	97.78%	0.97	100.00%	1.00
Min	26.67%	0.06	64.44%	0.61	26.70%	0.06	26.67%	0.06	95.56%	0.91
Avg	75.35%	0.72	85.45%	0.86	83.62%	0.79	85.46%	0.81	97.48%	0.97

## Data Availability

Not applicable.

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
