# Peer review of "Rapid Identification of Wild *Gentiana Genus* in Different Geographical Locations Based on FT-IR and an Improved Neural Network Structure Double-Net"

_molecules, 2022, doi:10.3390/molecules27185979_

Round 1
Reviewer 1 Report
· The new method of deep learning “Double Net” has shown a very high rate of accuracy in separating Gentiana species compared to the other methods. This nondestructive method of origin identification could initiate a new start for other valuable herbs as well as other valuable tree species for certification purposes.
· In the reference, after 19, instead of 20, there is 1 (Pei, Y.F.; Zuo, Z.T.; Zhang, Q.Z.; Wang, Y.Z.).
· In the introduction “Random Random Forest (RF)” needs correction (Ref: 1 Pei, Y.F.; Zuo, Z.T.; Zhang, Q.Z.; Wang, Y.Z.).
· Scientific names should be italic (Paris polyphylla)
· In section 2.4.1 there is a repetition (the highest accuracy of SVM model for wild Gentiana genus origin classification result is).
· Ref. 37 mentioned in the text but not listed in the reference section.
· Ref. 31 does not refer to the content mentioned.
· In section 2.2 reference is required for the assignment of different spectral bands in FTIR spectroscopy.
Author Response
Dear Reviewer 1:
Thank you for your comments concerning our manuscript entitled “Rapid identification of wild Gentiana genus in different geographical locations based on FT-IR and an improved neural network structure Double-Net” (ID: molecules-1890618). Those comments are all valuable and very helpful for revising and improving our paper, as well as the important guiding significance to our research. We have studied the comments carefully and made a correction that we hope to meet with your approval. Revised portions are marked in red in the paper. The main corrections in the paper and the responses to your comments are included in the attached PDF.
We appreciate for your warm work earnestly, and hope that the correction will meet with approval. Once again, thank you very much for your comments and suggestions.

Reviewer 2 Report
The description of the method needs to be revised.
The demarcation to the state of the art is not quite clear to me.
Other questions:
where were the difficulties in analysis and evaluation?
Where are the applications of this method?
Author Response
Dear Reviewer 2:
Thank you for your comments concerning our manuscript entitled “Rapid identification of wild Gentiana genus in different geographical locations based on FT-IR and an improved neural network structure Double-Net” (ID: molecules-1890618). Those comments are all valuable and very helpful for revising and improving our paper, as well as the important guiding significance to our research. We have studied the comments carefully and made a correction that we hope to meet with your approval. Revised portions are marked in red in the paper. And the main corrections in the paper and the responses to your comments are included in the attached PDF.
We appreciate for your warm work earnestly, and hope that the correction will meet with approval. Once again, thank you very much for your comments and suggestions.

Reviewer 3 Report
Manuscript entitled " Rapid identification of wild Gentiana genus in different geographical locations based on FT-IR and an improved neural network structure Double-Net” can be considered for publication in the journal “Molecules”, after the minor revision.
My specific comments are:
1. Authors need to mention the rationale to perform artificial neural network (ANN) and whether can it be used in clinical analysis?
2. Different analytical techniques that can also be used for the identification of Gentiana genus should be mention in the introduction section of the manuscript. Incorporate UPLC, HPLC and NMR based following references.
i. Sasaki, Nobuhiro, Keiichirou Nemoto, Yuzo Nishizaki, Naoki Sugimoto, Keisuke Tasaki, Aiko Watanabe, Fumina Goto et al. "Identification and characterization of xanthone biosynthetic genes contributing to the vivid red coloration of red‐flowered gentian." The Plant Journal 107, no. 6 (2021): 1711-1723.
ii. Pan, Zheng, Feng Xiong, Yi-Long Chen, Guo-Guo Wan, Yi Zhang, Zhi-Wei Chen, Wen-Fu Cao, and Guo-Ying Zhou. "Traceability of geographical origin in Gentiana straminea by UPLC-Q exactive mass and multivariate analyses." Molecules 24, no. 24 (2019): 4478.
iii. Khalil, Adila, and Mohammad Kashif. "Nuclear Magnetic Resonance Spectroscopy for Quantitative Analysis: A Review for Its Application in the Chemical, Pharmaceutical and Medicinal Domains." Critical Reviews in Analytical Chemistry (2021): 1-15.
3. In the section 2.1.2, Results of PCA processing, “the first 15 principal components have the largest contribution and the last components have almost zero” this is not easily validated by the Figure 2. Please rectify the figure (s) scale for more clarity.
4. Please include the composition of the Gentiana genus so to verify the peaks of FTIR. In the section 4.1 Sample Preparation, authors have written that they have washed the root samples with tap water. Why not distilled water?
5. The grammatical and typographical errors within the manuscript should be rectified.
Author Response
Dear Reviewer 3:
Thank you for your comments concerning our manuscript entitled “Rapid identification of wild Gentiana genus in different geographical locations based on FT-IR and an improved neural network structure Double-Net” (ID: molecules-1890618). Those comments are all valuable and very helpful for revising and improving our paper, as well as the important guiding significance to our research. We have studied the comments carefully and made a correction that we hope to meet with your approval. Revised portions are marked in red in the paper. The main corrections in the paper and the responses to your comments are included in the attached PDF.
We appreciate for your warm work earnestly, and hope that the correction will meet with approval. Once again, thank you very much for your comments and suggestions.

Round 2
Reviewer 3 Report
The authors have successfully addressed all the comments and incorporated all the suggestions in the revised manuscript.